# Your Vision-Language Model Can't Even Count to 20: Exposing the Failures of VLMs in Compositional Counting

## Abstract

Vision-Language Models (VLMs) have become a central focus of today's AI community, owing to their impressive abilities gained from training on large-scale vision-language data from the Web. These models have demonstrated strong performance across diverse tasks, including image understanding, video understanding, complex visual reasoning, and embodied AI. Despite these noteworthy successes, a fundamental question remains: Can VLMs count objects correctly? In this paper, we introduce a simple yet effective benchmark, **VLMCountBench**, designed under a minimalist setting with only basic geometric shapes (e.g., triangles, circles) and their compositions, focusing exclusively on counting tasks without interference from other factors. We adopt strict independent variable control and systematically study the effects of simple properties such as color and size in a controlled ablation. Our empirical results reveal that while VLMs can count reliably when only one shape type is present, they exhibit substantial failures when multiple shape types are combined (i.e., compositional counting). This highlights a fundamental empirical limitation of current VLMs and motivates important directions for future research.

## 1 Introduction

Vision-Language Models (VLMs) have recently emerged as one of the most influential paradigms in artificial intelligence (Gemma, 2025; Wang et al., 2024a; OpenAI, 2024). By jointly training on large-scale paired data from the web, VLMs have demonstrated impressive generalization across a wide range of tasks, including image captioning, video understanding, visual question answering, visual reasoning, and embodied AI (Driess et al., 2023; Cheng et al., 2024; Wang et al., 2024b). These models form the foundation of many recent multimodal systems and are increasingly deployed in real-world applications. Their ability to align vision and language representations in a unified framework has positioned them as a strong foundation for multimodal research and practice.

Despite these remarkable successes, a fundamental question persists: Do VLMs possess reliable basic perceptual abilities? Among these, counting plays a central role, as it underlies numerous higher-level reasoning skills and everyday applications. Counting is both a simple and fundamental visual task that requires identifying discrete objects and enumerating them accurately. Prior work has already raised concerns in related domains. Generative models, for instance, often fail to produce the correct number of objects in synthetic images (Petsiuk et al., 2022; Cao et al., 2025; Hui et al., 2024) and videos (Guo et al., 2025; Sun et al., 2025), and CLIP-based models have been shown to struggle with distinguishing and enumerating multiple objects in classification and retrieval settings (Jiang et al., 2023; Paiss et al., 2023; Zhang et al., 2024). However, the specific counting ability of VLMs remains less systematically explored. This motivates our research question:

**Question 1.** *Can state-of-the-art VLMs reliably perform simple and compositional counting tasks?*

While some existing benchmarks touch on VLMs' ability to count, they typically do so in complex or noisy environments (Li et al., 2024a;b; Xu et al., 2024). For example, datasets designed for visual question answering or captioning may contain counting-related queries, but these are embedded within broader tasks involving recognition, commonsense reasoning, or natural image understanding. As a result, it is difficult to disentangle whether a model's failure arises from counting itself

or from unrelated challenges. Similarly, large-scale natural image benchmarks (e.g., COCO (Lin et al., 2014) object detection dataset with labels on the quantity of objects) introduce uncontrolled variability, making it nearly impossible to isolate the exact conditions that cause performance degradation. Thus, despite progress, there remains no controlled and minimalist benchmark dedicated specifically to testing counting in VLMs.

To address this gap, we introduce **VLMCountBench**, a benchmark designed under a strictly minimalist setting. The benchmark consists of simple geometric shapes (e.g., triangles, circles) and their compositions, thereby removing semantic complexity and focusing exclusively on counting. This setting allows us to implement precise variable control, systematically manipulating factors such as color, size, rotation, and overlap. By conducting ablation studies under these conditions, we can rigorously analyze VLM performance and identify the specific challenges that lead to counting failures.

We carry out a comprehensive empirical evaluation across multiple state-of-the-art VLMs (), covering both open-source and commercial private models, focusing on both single-shape and multi-shape settings. Our results reveal several striking findings:

- VLMs can count reliably when only a single shape type is present, achieving high accuracy in simple counting scenarios.
- VLMs exhibit substantial failures in **compositional counting**, where two or more shape types coexist. These failures persist even when the task involves small numbers of objects and minimal visual complexity.
- Performance deteriorates consistently across variations in color, size, rotation, and overlapping, indicating a lack of stability to simple visual properties.

**Roadmap.** In Section 2, we review the related works. In Section 3, we present our proposed benchmark. In Section 4, we present the main experimental results. We introduce the prompt refinement in Section 5. In Section 6, we conclude our paper.

## 2 RELATED WORKS

**Vision-Language Models.** Motivated by the impressive success of Large language models (LLMs) (Brown et al., 2020; Wei et al., 2022; Touvron et al., 2023; Chung et al., 2024), scholarly attention is progressively shifting toward the exploration and development of vision-language models, as they have the potential to connect vision and language, achieve more natural human-computer interaction (Kim et al., 2025b), and advance tasks such as visual question answering (Lin et al., 2023; Kim et al., 2025a) and multimodal reasoning (Lee et al., 2024a; Chia et al., 2024). One significant leap in this area is the revolutionary Visual ChatGPT (Wu et al., 2023), which combines the reasoning ability of language models with several visual models to achieve natural language-driven image generation, editing, and understanding. Besides, PaLM-E (Driess et al., 2023) has effectively integrated text and vision, achieving remarkable results across a variety of tasks (Xu et al., 2016; Marino et al., 2019). Flamingo (Alayrac et al., 2022) integrates frozen large language models with visual encoders through cross-attention layers, achieving few-shot learning for visual language tasks. Conversely, BLIP2 (Li et al., 2023) effectively connects frozen Large Language Models (LLMs) with visual input through a lightweight Q-Former module, which converts image features into a format that LLMs can understand. This design enables high performance in various tasks with minimal additional training. Well-known models such as InstructBLIP (Marino et al., 2019) and LLaVA (Liu et al., 2023) have significantly advanced the field by introducing diverse visual instruction-tuning datasets. While prior vision-language models have demonstrated impressive performance across diverse multimodal tasks, their ability to perform precise quantitative analysis on images remains largely unexplored. To address this gap, we propose VLMCountBench to offer insights into their numerical understanding in visual scenes.

**Benchmarks for Vision-Language Models.** With the rapid development of Vision-Language Models (VLMs), researchers designed some benchmarks such as TextVQA (Singh et al., 2019), GQA (Hudson & Manning, 2019), and DocVQA (Mathew et al., 2021) to evaluate the ability of VLMs on individual tasks. However, while these task-specific benchmarks provide valuable insights, they do not fully reflect the overall capabilities of VLMs in real-world applications. There-

fore, recent efforts (Huang et al., 2024; Yue et al., 2024; Das et al., 2024) have shifted toward developing more comprehensive evaluation benchmarks. Meanwhile, VHELM (Lee et al., 2024b) comprehensively evaluates the performance of VLMs in multiple dimensions such as perception, reasoning, multilingual ability, and robustness. In addition, several representative benchmarks have been proposed to target different aspects of multimodal evaluation. For example, Perception Test (Patraucean et al., 2023) focuses on measuring fine-grained perceptual capacity such as color, shape, and size. LVLM eHub (Xu et al., 2024) combines multiple comprehensive benchmarks to design an evaluation platform that covers a wide range of multimodal tasks. LLaVA Bench (Liu et al., 2023), LAMM (Yin et al., 2023), and Touchstone (Bai et al., 2023) leverage GPT-based evaluators to assess model outputs, thereby reducing potential biases introduced by human annotators. Beyond general-purpose benchmarks, some works focus on constructing targeted datasets for more objective and fine-grained evaluation of VLM. MME (Chaoyou et al., 2023) and MMBench (Liu et al., 2024) are designed to strengthen the objective evaluation of VLMs by introducing 2,194 true/false questions and 2,974 multiple-choice questions across diverse ability dimensions. Although existing benchmarks effectively evaluate various VLM capabilities, they primarily target concrete visual entities (e.g., objects, scenes) and largely ignore numerical counting in visual contexts, which motivates the creation of **VLMCountBench**.

## 3 BENCHMARK

In Section 3.1, we introduce the evaluated models in this benchmark. In Section 3.2, we present the prompts to evaluate

### 3.1 EVALUTAED MODELS

Table 1: **Key Details of the Large Vision-Language Models.** Gemini-2.5 is a closed-source model that does not provide any information about its parameters.

| Model | Source | Year | # Output Tokens | # Params |
|---|---|---|---|---|
| Gemini 2.5 Flash | (Comanici et al., 2025) | 2025 | 64k | N/A |
| GPT-4o | (OpenAI, 2024) | 2024 | 16K | 200B |
| Ernie 4.5 | (Baidu, 2025) | 2025 | 16k | 47B |
| GLM 4.5v | (Hong et al., 2025) | 2025 | 64k | 12B |
| Gemma 3 27B | (Gemma, 2025) | 2025 | 128k | 27B |
| Qwen 2.5 72B | (Yang et al., 2025) | 2025 | 32K | 72B |
| Kimi VL A3B | (Du et al., 2025) | 2024 | 32K | 3B |
| Llama 4 Maverick | (Meta, 2025) | 2025 | 4K | 17B |

We evaluate eight state-of-the-art language models via the OpenRouter API, using their default context lengths and provider settings without any manual adjustment. All inference runs were performed without chain-of-thought prompting; however, Kimi VL A3B (Du et al., 2025) and Llama 4 Maverick (Meta, 2025) inherently expose chain-of-thought style reasoning that cannot be disabled, so any intermediate reasoning was ignored and only final outputs were considered.

**Open-source models**. Gemma 3 27B (Gemma, 2025) and Qwen 2.5 72B (Yang et al., 2025) provide long-context handling (default capacities of roughly 128 k and 32 k tokens respectively) and support high-resolution images where applicable. Kimi VL A3B (Du et al., 2025), a lightweight 3B parameter vision-language model, and Llama 4 Maverick (Meta, 2025), a 17B parameter text-focused model with a 4k token window, represent smaller, more agile configurations. Ernie 4.5 47B (Baidu, 2025) and GLM 4.5v 12 B (Hong et al., 2025) extend open-source multimodal capabilities with default 16 k and 64 k generation limits, respectively, and adhere to the common image side maximum of 1024 px established by their providers.

**Closed-source models**. Gemini 2.5 Flash (Comanici et al., 2025), from Google DeepMind, is optimized for fast multimodal inference with a default 64k token limit and image handling up to 1024 px. GPT-4o (OpenAI, 2024), OpenAI's flagship multimodal system with around 200B parameters, operates under a 16k token default and similar image size constraints.

For all models open and closed, we did not modify decoding hyperparameters or preset any structured outputs beyond provider defaults, ensuring a consistent evaluation setting across architectures and access modalities.

## 3.2 BENCHMARK PROMPTS AND INPUT IMAGES

Our benchmark is designed to directly evaluate the basic counting ability of vision-language models (VLMs), while minimizing the influence of confounding factors such as complex scene understanding or higher-level reasoning. We adopt a deliberately simple setting where the task is restricted to counting a small number of basic geometric shapes. This allows us to isolate and probe the fundamental ability of VLMs to perform object counting. Despite the simplicity of this setting, we will show that VLMs still exhibit significant failures. The benchmark considers three object types, triangle, square, and circle, and three levels of composition: one, two, or three object types in the same image. For each image, the quantity of objects is sampled between 1 and 20. An illustration of our benchmark is shown in Figure 1.

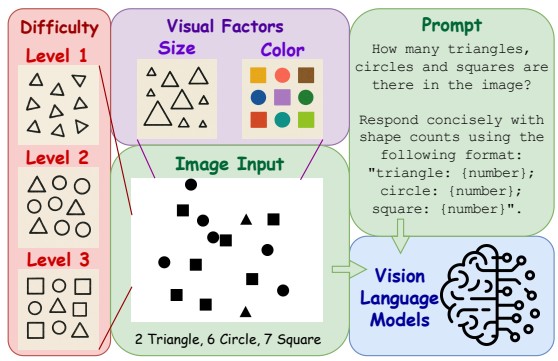

Figure 1: Our experimental design to let VLMs perform object counting.

**Prompts.** To construct queries, we combine three basic concepts: <object>, <level> of composition, and <quantity>. The options are:

- <object>: 'triangle', 'square', 'circle'.

- <level>: 'level 1', 'level 2', 'level 3'.

- <quantity>: 1, 2, 3, ..., 20.

In 'level 1', the image contains only one type of object. In 'level 2', two types of objects are present, and in 'level 3', three different types of objects are shown. The corresponding prompt templates are given below:

---

**Level 1 Prompt Template $P_1$**

How many <object 1> are there in the image?
Respond concisely with shape counts using the following format: "<object 1>: {number}". For example: "<object 1>: 7". The number 7 is provided as an example only and does not represent the actual quantity of objects in the image.
[image: <quantity 1> of <object 1>]

---

**Level 2 Prompt Template $P_2$**

How many <object 1> and <object 2> are there in the image?
Respond concisely with shape counts using the following format: "<object 1>: {number}; <object 2>: {number}". For example: "<object 1>: 9; <object 2>: 13". The numbers 7 and 13 are provided as examples only and do not represent the actual quantity of objects in the image.
[image: <quantity 1> of <object 1>, <quantity 2> of <object 2>]

---

---

**Level 3 Prompt Template $P_3$**

How many <object 1>, <object 2> and <object 3> are there in the image?
Respond concisely with shape counts using the following format: "<object 1>: {number}; <object 2>: {number}; <object 3>: {number}". For example: "<object 1>: 3; <object 2>: 11; <object 3>: 6". The numbers 3, 11, and 6 are provided as examples only and do not represent the actual quantity of objects in the image.
[image: <quantity 1> of <object 1>, <quantity 2> of <object 2>, <quantity 3> of <object 3>]

---

Here, [image: ...] denotes the actual input image containing the specified objects. The placeholders <object 1>, <object 2> and <object 3> always correspond to distinct object types (e.g., a query may ask about triangles and squares, but never triangles and triangles).

An example prompt at 'level 2' is shown below:

---

**Prompt Example 1**

How many `triangles` and `circles` are there in the image?
Respond concisely with shape counts using the following format: "`triangles`: {number}; `circles`: {number}". For example: "`triangles`: 9; `circles`: 13". The numbers 9 and 13 are provided as examples only and do not represent the actual quantity of objects in the image.
[image: 7 `triangles`, 15 `circles`]

---

For each level, we randomly sample from all possible combinations of objects and quantities, and retain 200 prompts per level. All images are generated automatically.

**Input Images.** To generate large-scale annotated data, we employ a simple automatic image generator. This can be implemented with basic Python commands, without relying on costly or time-consuming modern generative models, while still being sufficient to reveal the counting limitations of VLMs. Each image is a $640 \times 480$ canvas with a white background and stored as a JPG file. All shapes are drawn with black borders, white interiors, identical sizes, and no rotation. They are placed uniformly at random on the canvas, with no overlaps, ensuring that object counts remain unambiguous and easily verifiable.

In the base setting, we restrict our benchmark to varying only quantity and composition. More complex properties that may affect counting performance, such as size, color, and overlapping, are deferred to the ablation study.

### 3.3 EVALUATION METRICS

For each test sample in our benchmark, we use two evaluation metrics: accuracy and relative error. Accuracy measures whether the VLM's response is exactly correct, while relative error provides a finer-grained evaluation by quantifying how far the prediction deviates from the ground truth. Let a single test sample be denoted as $q := (p, x, y, m)$, where $p$ is the input prompt, $x$ is the input image, $y \in \mathbb{N}_+^m$ is the ground-truth vector of object counts, and $m \in \{1, 2, 3\}$ is the number of object types. For example, in Prompt Example 1 with two object types (triangle and circle) and counts 7 and 13, we have $y = [7, 13]^\top$ and $m = 2$.

**Accuracy.** Accuracy evaluates whether the prediction matches the ground truth for each object type. Let $\mathcal{Q}$ denote the set of test samples of interest (e.g., all 'Level 2' samples). The metric is defined as:

$$\text{Accuracy}(\mathcal{Q}) := m^{-1}|\mathcal{Q}|^{-1} \sum_{(p,x,y,m)\in\mathcal{Q}} \sum_{i=1}^{m} \mathbf{1}[\text{VLM}(p,x)_i = y_i],$$

where $\mathbf{1}[\cdot]$ is the indicator function, which returns 1 if the condition inside is true and 0 otherwise, and $\text{VLM}(p, x) \in \mathbb{N}_+^m$ is the predicted object counts.

Intuitively, for each sample $q$, we compute the fraction of object types predicted exactly correctly, then average over all samples in $\mathcal{Q}$. For instance, if an image contains three object types (triangle, circle, square) and the model predicts only the square count correctly, then the accuracy for this sample is $1/3$. The final accuracy is the mean of such values over all test samples.

**Relative Error.** While accuracy captures exact correctness, it does not reflect how close the prediction is when incorrect. To address this, we use relative error, which measures the normalized deviation of predicted counts from ground truth. Formally:

$$\text{RelativeError}(\mathcal{Q}) := m^{-1}|\mathcal{Q}|^{-1} \sum_{(p,x,y,m)\in\mathcal{Q}} \sum_{i=1}^{m} y_i^{-1} \cdot |\text{VLM}(p,x)_i - y_i|,$$

where $\text{VLM}(p,x) \in \mathbb{N}_+^m$ again denotes the predicted counts.

This metric computes, for each sample $q$, the average relative error across object types, and then averages over all samples in $\mathcal{Q}$. For example, if an image contains 16 circles and 10 squares, and the model predicts 8 circles and 8 squares, then the relative error is: $0.5 \cdot (|8-16|/16 + |8-10|/10) = 0.5 \cdot (0.5+0.2) = 0.35$. Thus, relative error provides a more detailed measure of how far predictions deviate from the true counts.

## 4 EXPERIMENTS

We present the main experimental results of the VLMCountBench in this section.

### 4.1 COMPOSITIONAL COUNTING

Table 2: **Overall Counting Accuracy and Relative Error Across various Object Types.** The models are listed in a sequence of descending overall count accuracy. We highlight the top 3 models with the best counting accuracy in blue, and top 3 models with the least relative error in red.

| Model | Level 1 | | Level 2 | | Level 3 | | Overall | |
|---|---|---|---|---|---|---|---|---|
| | Count Acc | Relative Error | Count Acc | Relative Error | Count Acc | Relative Error | Count Acc | Relative Error |
| Gemma3 27B | 0.26 | 0.14 | 0.21 | 0.23 | 0.22 | 0.25 | 0.23 | 0.21 |
| Kimi VL A3B | 0.29 | 0.23 | 0.22 | 0.27 | 0.19 | 0.30 | 0.23 | 0.27 |
| Llama4 Maverick | 0.38 | 0.15 | 0.33 | 0.14 | 0.25 | 0.19 | 0.32 | 0.16 |
| Gpt-4o | 0.44 | 0.07 | 0.39 | 0.10 | 0.23 | 0.17 | 0.35 | 0.11 |
| Ernie 4.5 | 0.52 | 0.05 | 0.43 | 0.08 | 0.38 | 0.10 | 0.44 | 0.08 |
| Gemini 2.5 Flash | 0.58 | 0.04 | 0.54 | 0.05 | 0.30 | 0.13 | 0.47 | 0.07 |
| GLM4.5v | 0.56 | 0.05 | 0.49 | 0.07 | 0.43 | 0.08 | 0.49 | 0.07 |
| Qwen2.5 72B | 0.60 | 0.04 | 0.56 | 0.05 | 0.45 | 0.07 | 0.53 | 0.05 |

We conduct experiments across three levels: contexts containing one object, two objects, and three objects. For each level, the number of shapes ranges from 1 to 20. Table 2 presents vision-language models' counting performance when varying both the number of object types (one, two, or three) and the number of object instances (ranging from 1 to 20) within the input context.

As shown in Table 2, current vision-language models still face significant challenges in counting, especially when dealing with multiple objects or diverse object types within the input images. Notably, even the best-performing vision-language model in our benchmark achieves only modest accuracy. For instance, Qwen2.5 72B (Yang et al., 2025) achieved an accuracy of 0.60 at Level 1, but its accuracy substantially declined to 0.45 at Level 3, highlighting the difficulty of the counting task. These findings point to the following insight:

**Observation 4.1.** *Our results reveal that current vision-language models do not perform ideally on the counting task, and there remains a substantial gap between existing vision-language models' capabilities and the reliable counting ability required for practical applications.*

Across all vision-language models in our benchmark, there is a refined relationship between accuracy and relative error, with relative error serving as a fine-grained metric specifically designed to evaluate counting performance. Even when a model's prediction is incorrect, a smaller relative error indicates that the predicted counts are closer to the ground truth. In addition, we observed that higher accuracy typically corresponds to smaller relative errors, indicating that models with higher accuracy tend to produce more reliable counting results. For example, Qwen2.5 72B (Yang et al., 2025) has the highest overall counting accuracy at 0.53 and the lowest overall relative error at 0.05. At Level 1, its accuracy is 0.60 with a relative error of 0.04, while at Level 3, the accuracy drops to 0.45 with a slight increase in relative error to 0.07, its relatively small relative error indicates that its counting results are usually close to ground truth, compared to models with lower accuracy and

larger relative errors, such as Kimi VL A3B (Du et al., 2025), which has an overall accuracy of 0.23 and a relative error of 0.27, demonstrating a certain degree of counting ability. This brings us a novel insight:

**Observation 4.2.** *Vision-language models that achieve higher accuracy tend to have smaller relative errors, indicating a stronger counting ability. Conversely, vision-language models with lower accuracy generally show larger relative errors, suggesting limited counting competence. This demonstrates that some vision-language models possess a certain degree of visual counting capability, while others struggle to reliably quantify objects.*

When the number of object types in the input image increases, we observe a clear trend: higher composition levels lead to reduced counting accuracy and increased relative error. For example, Gemini 2.5 Flash achieves a counting accuracy of 0.58 at Level 1, which decreases to 0.54 at Level 2 and further drops to 0.30 at Level 3. Its relative error correspondingly rises from 0.04 to 0.05and then to 0.13. Similar phenomena are observed in GLM4.5v and Qwen2.5 72B, where accuracy declines and relative error rises as more object types are present. From this, we derive the following insight:

**Observation 4.3.** *Even one of the best-performing models experiences substantial performance degradation as the scene composition becomes more complex. This indicates that current vision-language models may struggle to distinguish multiple object types in a single visual scene, and the interaction between object types (e.g., similar appearances) may further confuse the vision-language models.*

## 4.2 IMPACT OF VISUAL PERTURBATIONS

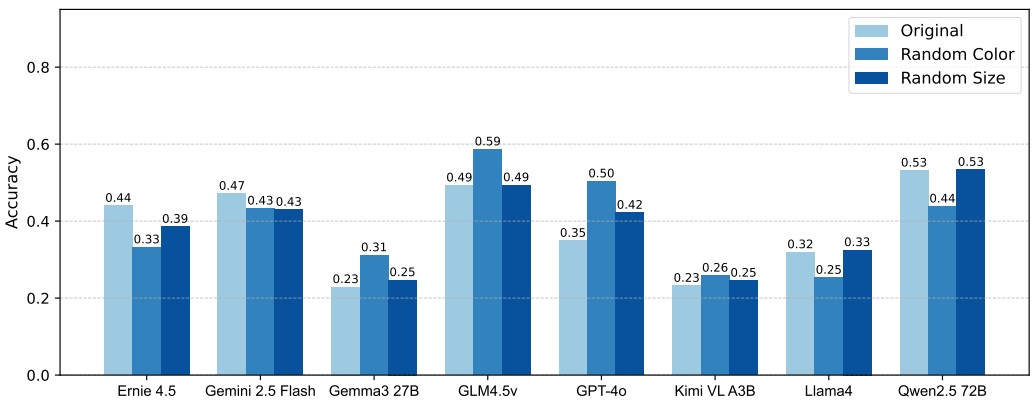

Figure 2: **Impact of Visual Perturbations on Model Accuracy**.

To better explore the current vision-language models' performance in the counting task. We conduct an ablation study based on our benchmark, VLMCountBench. Figures 2 report counting performance, measured by accuracy and relative error, under the three ablation settings. In the original setting, which serves as our main experiment, all shapes are uniform in size and colored black. In the random color setting, shapes are randomly assigned different colors while all other conditions remain identical to the main experiment. In the random size setting, the shape will randomly resize, possibly larger or smaller, with all other conditions remaining unchanged. This setting enables us to systematically evaluate the impact of visual perturbations, such as color and size variations, on the counting performance of vision-language models.

As illustrated in Figure 2, applying random color and random size perturbations to input images leads to varying impacts on counting performance across vision-language models. In particular, GLM4.5v (Hong et al., 2025) and GPT-4o (OpenAI, 2024) actually benefit from color variations, showing notable increases in accuracy compared with the original setting, possibly because the color differences make objects easier to distinguish. while Ernie 4.5 (Baidu, 2025) and Qwen2.5 72B (Yang et al., 2025) experience substantial drops, suggesting that these models may rely on specific color distributions learned during training, and that color randomization can disrupt their

counting mechanism. In contrast, size perturbations generally cause smaller impacts on performance. Qwen2.5 72B (Yang et al., 2025) and GLM4.5v (Hong et al., 2025) remain relatively high accuracy, while Gemma3 27B (Gemma, 2025) and Kimi VL A3B (Du et al., 2025) continue to perform at lower levels. Based on the above analysis, we make the following observations:

**Observation 4.4.** *Perturbations in color and size could positively or negatively affect counting performance, and the majority of vision-language models are more sensitive to color changes than to size variations, reflecting the different robustness features between vision-language models.*

## 5 Prompt Refinement

In this subsection, we evaluate whether the counting limitations of VLMs can be simply resolved by prompt refinements. In Section 5.1, we illustrate the prompt refinement in our work. We present the prompt refinement result and discuss the current discoveries regarding the counting capability of VLMs in Section 5.2.

### 5.1 The Proposed Prompts

Let the prompt template for the three difficulty levels in Section 3.2 be $P_1, P_2, P_3$. In this section, we introduce several refinement prompts that hint the VLMs to solve the complex counting task by task decomposition, splitting the original task into smaller and manageable parts. These refinement prompts are denoted as $P_{r,1}$ and $P_{r,2}$, and our final prompt used to evaluate the VLMs is denoted by $P \parallel P_r$, where $\parallel$ represents concatenation.

Specifically, $P_r$ has several instantiations.

**Spatial Decomposition.** We found that directly requiring the VLMs to provide a global number may result in omissions or duplications in image counting tasks. Inspired by this, we designed a spatial decomposition approach that breaks down counting tasks into spatial dimensions. We demand VLMs first count the number of objects in the left half of the image, then count the right half, and finally add the results of the two parts. We believe that such prompt refinement can help the VLMs form a local-global inference process, thereby improving the counting performance. Our prompt can be shown as follows:

> **Spatial Decomposition Prompt $P_{r,1}$**
>
> First count the objects on the left half of the image, then the right half, and add them together.

In specific applications, such as counting triangles and circles in an image, we require the VLMs to "count the left first, then the right, and finally merge the results", and output the quantities of each category in a fixed format. The details example can be shown as follows:

> **A Level 2 Spatial Decomposition Example $P_2 \parallel P_{r,1}$**
>
> How many `triangles` and `circles` are there in the image?
> Respond concisely with shape counts using the following format: "`triangles`: {number}; `circles`: {number}". For example: "`triangles`: 9; `circles`: 13". The numbers 9 and 13 are provided as examples only and do not represent the actual quantity of objects in the image.
> First count the objects on the left half of the image, then the right half, and add them together.
> [image: 7 `triangles`, 15 `circles`]

**Type Decomposition.** Another human-inspired method for counting a great number of objects in an image is to first count one category of objects and then proceed to the next. The type decomposition strategy of counting by category could avoid confusion between different categories and improve the counting performance of the VLMs. We define our prompt as follows:

---

**Type Decomposition Prompt $P_{r,2}$**

Count all instances of <object 1>first, then all instances of <object 2>, and then all instances of <object 3>.

---

For example, when the image contains triangles, circles, and squares, we explicitly require the VLMs to "count triangles first, then circles, and finally squares", and provide the results in a unified format. The details example can be shown as follows:

---

**A Level 3 Spatial Decomposition Example $P_3 \parallel P_{r,2}$**

How many `triangles`, `circles`, and `squares` are there in the image?
Respond concisely with shape counts using the following format: "`triangles`: {number}; `circles`: {number}; `squares`: {number}". For example: "`triangles`: 9; `circles`: 13; `squares`: 6". The numbers 9, 13, and 6 are provided as examples only and do not represent the actual quantity of objects in the image.
Count all instances of `triangles` first, then all instances of `circles`, and then all instances of `squares`.
[image: 7 `triangles`, 15 `circles`, 10 `squares`]

---

## 5.2 RESULTS AND DISCUSSION

Table 3: **Counting Accuracy and Relative Error for Spatial and Type Decomposition.** The models are listed in a sequence of descending overall count accuracy. We highlight the top 3 models with the best counting accuracy in blue, and top 3 models with the least relative error in red.

| Model | Original | | Spatial | | Type | |
|---|---|---|---|---|---|---|
| | Count Acc | Relative Error | Count Acc | Relative Error | Count Acc | Relative Error |
| Gemma3 27B | 0.26 | 0.14 | 0.30 | 0.15 | 0.16 | 0.49 |
| Kimi VL A3B | 0.29 | 0.23 | 0.18 | 0.37 | 0.15 | 0.50 |
| Llama4 Maverick | 0.38 | 0.15 | 0.35 | 0.14 | 0.21 | 0.44 |
| Gpt-4o | 0.44 | 0.07 | 0.43 | 0.08 | 0.26 | 0.40 |
| Ernie 4.5 | 0.52 | 0.05 | 0.43 | 0.09 | 0.26 | 0.41 |
| Gemini 2.5 Flash | 0.58 | 0.04 | 0.46 | 0.07 | 0.29 | 0.39 |
| GLM4.5v | 0.56 | 0.05 | 0.46 | 0.08 | 0.31 | 0.39 |
| Qwen2.5 72B | 0.60 | 0.04 | 0.47 | 0.07 | 0.35 | 0.38 |

Table 2 presents the counting accuracy and relative error under different refinement strategies. The results demonstrate that compared to the original counting prompts, applying spatial decomposition prompts will slightly reduce accuracy and increase relative error. Although the decomposition strategy provides a more structured step-by-step counting process, additional decomposition steps may introduce errors or complicate the inference process, resulting in a slight decrease in counting performance. In contrast, type decomposition exhibits an even larger performance drop in both accuracy and relative error, demonstrating that for current VLMs, dividing by object type will introduce greater noise in the counting process.

## 6 CONCLUSION

In our study, we propose **VLMCountBench**, a novel benchmark specifically designed to evaluate the counting ability of vision-language models under controlled, minimalist settings. Through systematic experiments on a series of state-of-the-art vision-language models, we found that current vision-language models face significant difficulties in accurately calculating objects in input images, especially in compositional counting scenarios involving multiple object types with varying attributes, such as size and color. These results reveal the fundamental limitations of existing vision-language models and emphasize the necessity of future research to enhance robust counting capabilities. We hope that **VLMCountBench** can provide valuable experience for future researchers to develop more accurate and reliable vision-language models.

ETHIC STATEMENT

This paper does not involve human subjects, personally identifiable data, or sensitive applications. We do not foresee direct ethical risks. We follow the ICLR Code of Ethics and affirm that all aspects of this research comply with the principles of fairness, transparency, and integrity.

REPRODUCIBILITY STATEMENT

We ensure the reproducibility of our empirical findings. For all experiments, we describe the sources of the LLM models, datasets, evaluation metrics, and experiment setup in the main text. All prompt templates used are also provided to support the reproducibility of our results.

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

# Appendix

**Roadmap.** Section A shows the model details of ten baseline vision-language models. Section B present additonal experiments.

## A    MODEL DETAILS

We present further details of vision-language models in this section.

- **GPT 4o** (OpenAI, 2024): Created by the OpenAI in 2024, GPT-4o is a closed-source multimodal model. GPT 4o integrates visual and language processing into a unified architecture, enabling tasks such as image understanding, multimodal reasoning, and interactive dialogue. The model supports multimodal inputs, including text, images, and audio, and it can generate outputs across modalities at a breakneck speed based on the problem.

- **Gemma 3** (Gemma, 2025): Developed by Google DeepMind and released in 2025. Gemma 3 is an open-source vision-language model. It supports multimodal inputs, allowing users to combine text and images within a single prompt. It supports over 140 languages and includes built-in safety tools for filtering sensitive visual content.

- **Qwen2 VL 72B** (Wang et al., 2024a): Qwen VL 72B is an open-source vision-language model by Alibaba in 2024. It supports multimodal input, including text and images, capable of processing high-resolution images and performing fine-grained understanding.

- **Gemini 2.5 Flash** (Comanici et al., 2025): Developed by Google DeepMind in 2025, Gemini 2.5 Flash is a closed-source multimodal model that supports processing text, image, video, and audio inputs. Besides, the model has built-in thinking capabilities to observe its reasoning process during the generation process

- **ERNIE 4.5 VL** (Baidu, 2025): ERNIE 4.5 VL is an open-source vision-language model from Baidu in 2025. It can integrate and text and images, providing different modes of thinking and non-thinking, and support long contextual lengths

- **GLM 4.5V** (Hong et al., 2025): GLM 4.5V is an open-source vision-language model released by Zhipu AI in 2025. It is capable of processing multiple types of inputs, including text, images, and video, and it can handle long-context tasks up to 66K tokens with high efficiency and accuracy.

- **Kimi VL A3B** (Du et al., 2025): Kimi VL A3B is an open-source vision-language model released by Moonshot AI in 2025. It supports a wide range of multimodal inputs, including text, high-resolution images, short video clips, and optional OCR or GUI inputs. In addition, it supports advanced reasoning using a "thinking mode", including text-guided image editing and style conversion.

- **Llama 4 maverick** (Meta, 2025): Llama-4-maverick is an open-source vision-language model from Meta. It adopts a Mixture-of-Experts (MoE) architecture with 17B active parameters, enabling efficient support of multimodal input, including text and high-resolution images, and provides a 128K token context window.

We also present the pricing details of all the mdoels in Figure 4.

Table 4: **Key Details of the Large Vision-Language Models.** (Free models up to 1000 requests per day)

| Model | free access? | price/prompt | Token Price |
|---|---|---|---|
| Gemini 2.5 Flash | No | $0.004 | $0.30/M input $2.50/M output $1.238/K input imgs |
| GPT-4o | No | $0.005 | $5/M input $15/M output $7.225/K input imgs |
| ERNIE 4.5 | No | $0.0007 | $0.14/M input $0.56/M output |
| GLM 4.5V | No | $0.001 | $0.5/M input $1.8/M output |
| Gemma 3 27B | Yes | $0.00005 | $0.067/M input $0.267/M output |
| Qwen 2.5 72B | Yes | $0.0001 | $0.25/M input $0.75/M output |
| Kimi VL A3B | Yes | $0.0001 | $0.025/M input $0.1/M output |
| Llama 4 Maverick | Yes | $0.0003 | $0.15/M input $0.6/M output $0.668/K input imgs |

## B   ADDITIONAL EXPERIMENTS

Due to space constraints, Figure 3 has been moved here.

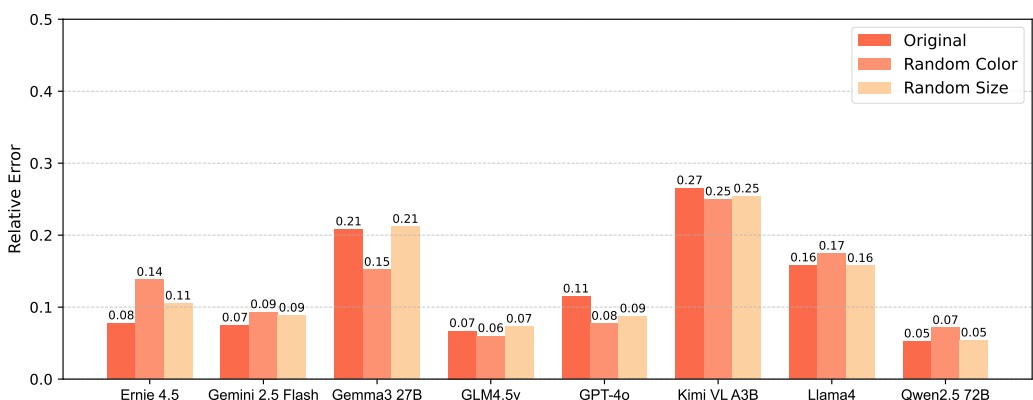

Figure 3: **Impact of Visual Perturbations on Model Relative Error**.

## LLM USAGE DISCLOSURE

LLMs were used only to polish language, such as grammar and wording. These models did not contribute to idea creation or writing, and the authors take full responsibility for this paper's content.

