# OpenReview forum: "Your Vision-Language Model Can’t Even Count to 20: Exposing the Failures of VLMs in Compositional Counting"
_ICLR.cc/2026/Conference — ICLR 2026 Conference Withdrawn Submission_

### Official Review · Reviewer_HArK · 2025-10-14

**Soundness:** 2
**Presentation:** 2
**Contribution:** 2
**Rating:** 4
**Confidence:** 3

**Summary:**

This paper investigates the counting ability of vision-language models in controlled synthetic scenes. The benchmark uses a small set of simple geometric shapes and manipulates visual factors such as color, size, and object interactions to probe when models over count or under count. The study reports that counting performance varies notably with visual conditions and that models can be misled by interactions or overlaps among objects. The experimental design is clear and the analyses are readable, but the scope is relatively narrow because the data are synthetic and restricted to three basic shapes. As a result, the generalization for real images and diverse categories is uncertain. The work is valuable as a diagnostic starting point, and it would benefit from broderr experiment, stronger prompt analysis.

**Strengths:**

1.The motivation is clear and the task has practical significance for reliable perception.

2.The finding that models are more sensitive to color changes than to size variations and that object interactions confuse counting is interesting and actionable for future work.

3.The synthetic design enables controlled ablations that isolate single factors without dataset noise.

**Weaknesses:**

1.Presentation can be improved. For Section 4, include an upfront overview of all experiments, and in the prompt refinement study report performance deltas against a defined baseline. Observation 4.2 reads wordy and could be tightened.

2.The prompt refinement investigation is largely heuristic. Some enhanced prompts reduce accuracy, so a deeper analysis of why prompts help or hurt would strengthen the claims.

3.The benchmark setting is relatively simple and narrow, leaving the generalizability of the measured counting capacity to real-world application unclear. A small extra study on real scenes or cross-dataset transfer would help.

4.Efficiency and robustness details are limited. Reporting effects of image resolution, number of image tokens and multiple seeds would clarify sensitivity.

- line 129 where a "." is missing.

**Questions:**

How does counting accuracy vary with inference settings beyond prompts, such as input image resolution and number of image tokens?

Do results persist across multiple random seeds and small perturbations of scene layout?

Is there evidence that performance on these synthetic compositions correlates with performance on real images?

---

### Official Review · Reviewer_yPTB · 2025-10-27

**Soundness:** 2
**Presentation:** 3
**Contribution:** 2
**Rating:** 2
**Confidence:** 4

**Summary:**

This work studies the ability of vision-language models (VLMs) to count the number of objects in an image. To do so the authors introduce a new benchmark called VLMCountBench that uses synthetically generated images of simple shapes of varying counts, colors, and layouts. The authors measure VLMs' ability to count the total number of objects when the same shape type versus multiple shape types are present. The authors show when VLMs are better at counting when the same shape type is present. The authors show VLMs also struggle with counting tasks when the color or sizes of the objects varies.

**Strengths:**

- The authors focus on counting, a fundmaental capability needed for visual understanding
- The main finding that LLMs struggle with compositional counting is interesting and points to an open-research problem for the community
- The authors rely on synthetically generated images to isolate the effect of external factors in their study and consider a reasonble set of leading vision-language models including both closed and open models.
- I also appreciate the additional experiments examining the effect of color, size, and layout.
- The overal presentation of the results and methods is clear

**Weaknesses:**

# Relation to existing work

The authors ought to do a much better job situating this contribution within the existing literature. The authors only briefly touch on general purpose benchmark in their related work overlooking many existing works that also study counting in vision-language. What exactly is missing from existing benchmarks that warrants this new VLMCountBench? For example, "CountBench" from  https://arxiv.org/abs/2302.12066 or  “Pascal-VOC Benchmark” “Emoji-Count” “Penguin Benchamrk” see  LVLM-Count https://arxiv.org/abs/2412.00686 as well as counting benchmarks studied in https://arxiv.org/abs/2408.04810. The authors should make a case for how VLMCountBench fits within the existing literature.

*Simulation versus real world counting*: the discussion above also leaves me wondering how do the findings on these simple synthetic shapes relate to real world counting tasks? Do we have evidence the counting capabilities measured using simple shapes correlates with more realistic counting tasks?

# Evaluation Methodology

*Counting a larger number of objects is intriscially more challenging*: do the authors control for the total number of objects across the three levels? For example, I'd be interested in seeing for a fixed number of total n objects, how does model performance differ when those objects are comprised of one, two, or three shapes. This isolates the natural difficulty in counting more objects from the models' capacity for compositional counting.

*What is the typical variation in counting accuracy?*: The authors claim  "even the best performing models experience substantial degradation as scene composition becomes more complex." While I do see the overall numbers are lower across levels, to substantiate this claim, the authors need to take into account the variability in each levels' counting accuracy. Are the differences within the expected margin of error?

How easy or difficult is it for models to perceive the objects in the scenes? Can you include more examples of the generated images used for evaluation for readers to get a sense of the difficulty of these counting tasks and the variations considered? I was only able to find a small set of examples in Figure 1.

# Overall Claim Justification


In the introduction the authors claim "VLMs can count reliably when only a single shape type is present."  However, according to Table 1, for most models around half of the time the models incorrectly count a single shape ("Level 1"), with the best model achieving only 60% accuracy. This claim, is central to the paper (to contrast against the lack of compositional counting skills), but does not seem well supported based on the evidence. It also is in contradiction with the bold title—"Can't count to 20" while "reliably counting a single shape" are claims that are inherently at odds. The authors should more carefully think about the overall contribution here.

While I do believe a systematic study of counting is valuable the authors need to do more work here to position the work and refine the central claims.

**Questions:**

To what extent is the deficiency in accuracy attributable to

1) a lack of models' ability to perceive the shapes in their various colors, layouts, and sizes? A baseline perception task to asses whether models reliable detect the shapes is an important baseline to isolate the counting capability of interest.

2) an inability to follow the instructions? In other words, how often does a model output a valid number when asked to count the number of objects or is the failure a result of the models' inability to answer the question in the requested format?

---

### Official Review · Reviewer_khua · 2025-10-31

**Soundness:** 4
**Presentation:** 4
**Contribution:** 2
**Rating:** 2
**Confidence:** 4

**Summary:**

This paper introduces VLMCountBench, a minimalist and controlled benchmark designed to test the counting ability of modern vision-language models (VLMs) such as GPT-4o, Gemini-2.5-Flash, Qwen-2.5-72B, Gemma-3-27B, and others. Unlike prior benchmarks (e.g., GQA, MMBench, LVLM-eHub), which mix counting with broader reasoning or perception challenges, this dataset isolates pure counting in simple geometric scenes containing 1–3 object types (triangle, circle, square) with quantities up to 20.

## Key Findings

The paper presents several key findings that collectively reveal fundamental weaknesses in modern VLMs when it comes to even the simplest forms of visual enumeration:

1. **VLMs fail at compositional counting**: authors show that VLMs can count relatively accurately when a single object type is present (Level 1), but their performance degrades sharply as the number of distinct object types increases—demonstrating severe failures in compositional counting.
2. **Counting failures persist under prompt refinements**: authors attempts to improve performance through prompt refinements—including spatial decomposition (“count left then right”) and type-wise decomposition (“count triangles first, then circles”)—either had negligible or negative effects, reinforcing that these failures are not due to prompt phrasing but stem from deeper architectural or representational limitations.
3. **Color and size perturbations affect models differently**: across all models, higher complexity leads to consistent increases in relative error, suggesting that their visual grounding collapses when faced with mixed categories. Even minor visual perturbations such as random changes in color or size significantly affect performance— implying unstable reliance on superficial cues rather than invariant object perception.

**Strengths:**

1. **Clear and minimalist benchmark design**: the use of simple geometric shapes under controlled conditions isolates counting ability without interference from semantic or contextual complexity.
2. **Broad model coverage**: evaluates both open-source (e.g., Qwen 2.5, Gemma 3, LLaMA 4) and closed-source (e.g., GPT-4o, Gemini 2.5) models, ensuring that conclusions generalize across architectures.

**Weaknesses:**

1. **novelty**: While the proposed VLMCountBench is clean and minimalist, its scope is relatively narrow and conceptually reminiscent of earlier synthetic frameworks such as CLEVR-Count, which also manipulates geometric shapes under controlled variations in color, size, and spatial composition. In contrast, CLEVR-Count operates in a 3D-rendered environment with lighting, occlusion, and viewpoint variation—making it more ecologically valid and visually diverse [1]. More recent works like [2] indeed uses CLEVR benchmark to demonstrate the limitation in VLMs on counting even with humans, showing the relative degradation in performance on these tasks. Compared to these, the present paper’s benchmark remains simplified to 2D flat shapes with uniform backgrounds, which limits generalization to naturalistic images or complex visual reasoning tasks. Although the minimalist approach offers interpretability, it reduces novelty relative to established compositional and counting testbeds that already probe similar perceptual challenges in more comprehensive domains.

2. **lack of architectural and representational analysis**: While the paper convincingly demonstrates that current VLMs fail at compositional counting, it stops short of exploring why these failures occur. Many prior works have already documented similar deficits in numerical reasoning and object individuation across models like CLIP, BLIP-2, and LLaVA. However, few studies, including this one, attempt to diagnose the root causes of these failures. The paper focuses primarily on performance-level comparisons (accuracy and relative error) but omits any architectural dissection or representational probing that could explain whether the bottleneck arises from the visual encoder’s limited spatial resolution, the cross-modal fusion mechanism’s loss of fine-grained token alignment, or the language model’s inability to represent discrete quantities.

[1] Johnson, J., Hariharan, B., Van Der Maaten, L., Fei-Fei, L., Lawrence Zitnick, C., & Girshick, R. (2017). Clevr: A diagnostic dataset for compositional language and elementary visual reasoning. In Proceedings of the IEEE conference on computer vision and pattern recognition (pp. 2901-2910).
[2] Nagar, A., Jaiswal, S., & Tan, C. (2024, June). Zero-shot visual reasoning by vision-language models: Benchmarking and analysis. In 2024 International Joint Conference on Neural Networks (IJCNN) (pp. 1-8). IEEE.

**Questions:**

1. **Error Categorization**:While the paper reports overall accuracy and relative error, it remains unclear what kinds of mistakes the models are actually making. A deeper error analysis could substantially strengthen the diagnostic value of the benchmark. For instance, are models primarily failing to count correctly while recognizing the right object types, or are they confusing categories altogether (e.g., labeling circles as triangles)? These two failure modes likely emerge from different underlying mechanisms—one from imprecise enumeration or attention drift within correctly localized regions, and the other from category binding errors in the cross-modal representation. Furthermore, it would be informative to know whether the errors exhibit systematic biases, such as consistent undercounting, overcounting, or confusion between visually similar shapes (e.g., triangles vs. squares). The authors could also analyze per-object-type accuracy to see whether certain shapes are inherently more difficult, or whether counting errors scale with object quantity (e.g., degradation past 5–7 items reflecting subitizing limits).
2. **Scaling of Complexity**: Another area that would benefit from deeper exploration is how performance scales with task complexity—specifically, with increasing numbers of objects, object types, and visual attributes. While the paper reports aggregate results for Level 1, Level 2, and Level 3 compositions, it does not analyze how accuracy and error evolve as the total count increases (e.g., from 1 to 20 objects). Such an analysis could reveal whether VLMs exhibit a threshold or saturation effect, similar to the human “subitizing” phenomenon, where precise counting is reliable only up to about 4–7 objects before errors sharply increase [1]. Observing such behavioral transitions could provide insights into whether current VLMs perform approximate numerosity estimation or rely purely on pattern-matching heuristics.

[1] Kaufman, E. L., Lord, M. W., Reese, T. W., & Volkmann, J. (1949). The discrimination of visual number. The American journal of psychology, 62(4), 498-525.

---

### Official Review · Reviewer_KTSB · 2025-11-01

**Soundness:** 2
**Presentation:** 2
**Contribution:** 1
**Rating:** 2
**Confidence:** 4

**Summary:**

The paper is quite interesting and the findings are very timely given the spread of VLMs both in academia and the industry. The scope is quite clear and there are enough details to reproduce the paper (although I didn't try it myself). Overall, I do feel that the limitations outweigh the benefits (see below) hereby I am leaning towards rejection unless the authors can address my concerns.

** Strength **
- very timely paper with a clear goal to stress test VLMs for a particular task
- overall the paper seems reproducible and there are enough details around experiments and setups
- The benchmark design and evaluation is thorough and sound

** Weakness **
- The contribution feels limited and redundant with prior work, e.g., https://arxiv.org/pdf/2408.04810 already looking at counting tasks and limitations of VLMs with further analysis
- No discussion or in-depth discussion to understand why the VLMs fail (e.g. which part of the architecture is failing)
- There are also a lot of typos that seem very easy to spot and make the submission feel rushed. For example Evalutaed, additonal, mdoels and so on

Overall, I feel that albeit sound, the contribution is extremely limited--more akin to an observation that is already quite known in the field--and quite redundant with prior work.

**Strengths:**

please see summary

**Weaknesses:**

please see summary

**Questions:**

please see summary

---

### Note · Authors · 2025-11-27

**Comment:**

We would like to sincerely thank all the reviewers for providing constructive feedback on our paper. We will carefully address these comments in our next version. After careful consideration, we have decided to withdraw this paper.

**Withdrawal Confirmation:**

I have read and agree with the venue's withdrawal policy on behalf of myself and my co-authors.